

# Carabid community structure in northern China grassland ecosystems: Effects of local habitat on species richness, species composition and functional diversity

Noelline Tsafack[1], François Rebaudo[2], Hui Wang[1], Dávid D. Nagy[3], Yingzhong Xie[1], Xinpu Wang[1] and Simone Fattorini[4]

[1] School of Agriculture, Ningxia University, Yinchuan, China
[2] UMR EGCE, IRD, University of Paris-Sud, CNRS, University of Paris-Saclay, Gif-sur-Yvette, France
[3] MTA-DE, Biodiversity and Ecosystem Services Research Group, Egyetem Sq. 1, Debrecen, Hungary
[4] Department of Life, Health and Environmental Sciences, University of L'Aquila, L'Aquila, Italy

Corresponding authors
Yingzhong Xie, yzh.xie@iCloud.com
Xinpu Wang,
wangxinpu@nxu.edu.cn

## ABSTRACT

**Background:** Most carabid beetles are particularly sensitive to local habitat characteristics. Although in China grasslands account for more than 40% of the national land, their biodiversity is still poorly known. The aim of this paper is to identify the main environmental characteristics influencing carabid diversity in different types of grassland in northern China.

**Methods:** We investigated the influence of vegetation (plant biomass, cover, density, height and species richness), soil (bulk density, above ground litter, moisture and temperature) and climate (humidity, precipitation and temperature) on carabid community structure (species richness, species composition and functional diversity—measured as body size, movement and total diversity) in three types of grasslands: desert, typical and meadow steppes. We used Canonical correspondence analysis to investigate the role of habitat characteristics on species composition and eigenvector spatial filtering to investigate the responses of species richness and functional diversities.

**Results:** We found that carabid community structure was strongly influenced by local habitat characteristics and particularly by climatic factors. Carabids in the desert steppe showed the lowest richness and functional diversities. Climate predictors (temperature, precipitation and humidity) had positive effects on carabid species richness at both regional and ecosystem levels, with difference among ecosystems. Plant diversity had a positive influence on carabid richness at the regional level. Soil compaction and temperature were negatively related to species richness at regional level. Climatic factors positively influenced functional diversities, whereas soil temperature had negative effects. Soil moisture and temperature were the most important drivers of species composition at regional level, whereas the relative importance of the various environmental parameters varied among ecosystems.

**Discussion:** Carabid responses to environmental characteristics varied among grassland types, which warns against generalizations and indicates that management programs should be considered at grassland scale. Carabid community structure is

strongly influenced by climatic factors, and can therefore be particularly sensitive to ongoing climate change.

# INTRODUCTION

Carabid (Coleoptera: Carabidae) assemblages are strongly influenced by habitat structure, especially as reflected by vegetation and soil characteristics (*Koivula et al., 1999*; *Brose, 2003*; *Taboada et al., 2008*), being particular sensitive to anthropogenic alterations (*Rainio & Niemelä, 2003*; *Koivula, 2011*). For these reasons, carabid distributional patterns and community structure can be strongly affected by land-use changes (*Eyre et al., 2003*; *Eyre & Luff, 2004*; *Kotze et al., 2011*; *Gobbi et al., 2015*; *Lyons et al., 2017*; *Lafage & Pétillon, 2016*).

For example, soil bulk density (SBD; an indicator of soil structure, also used to estimate soil compaction, *Rabot et al., 2018*), and soil moisture (SM), two environmental characteristics that are altered by human activities, are key factors for carabid ecology. *Kagawa & Maeto (2014)* showed that the abundance of some species is associated with different degrees of SM, while *Magura, Tóthmérész & Elek (2003)* reported that soil compaction negatively influences carabid activities, such as egg-laying and burrowing during aestivation and hibernation.

Carabids can also be impacted by the amount of litter on the soil (*Magura, Tóthmérész & Elek, 2003*) since it modulates both SM and soil temperature (ST; *Xiao et al., 2014*), improves soil fertility and increases food availability (*Koivula et al., 1999*; *Magura, Tóthmérész & Elek, 2005*). It has been observed that leaf litter increases the number of carabids by increasing habitat heterogeneity, producing favorable microsites and allowing a separated vertical distribution in the litter layer which may lead to decreased intra- and inter- species competition (*Magura, Tóthmérész & Elek, 2003*). Another soil characteristic that is particularly relevant for carabids is ST (*Hiramatsu & Usio, 2018*; *Robinson et al., 2018*) because this physical parameter influences various species-specific temperature-dependent performances (*Merrick & Smith, 2004*).

Vegetation composition and diversity also impact carabid community structure (*Koricheva et al., 2000*; *Brose, 2003*; *Schaffers et al., 2008*; *Zou et al., 2013*; *Pakeman & Stockan, 2014*; *Ng et al., 2018a*) and functional diversity (*Liu et al., 2014*; *Pakeman & Stockan, 2014*; *Spake et al., 2016*; *Magura, 2017*), because plants provide both shelter and food, directly (for herbivores) and indirectly (by providing prey for predators).

Within local environmental characteristics, in addition to vegetation and soil characteristics, climate plays an important role in carabid ecology. Temperature influences flight, speed of digestion, fecundity and also larval survival (*Thiele, 1977*; *Butterfield, 1996*; *Lövei & Sunderland, 1996*), and *Rainio & Niemelä (2003)* showed that ambient temperature and humidity were the two main abiotic factors influencing carabid species.

Carabid community structure may also be positively impacted by precipitation through the responses of plants to this factor. Because plant diversity and biomass increase with increasing precipitation (*Yan et al., 2015*), sites with more rainfall should provide more habitat diversity and food for carabids.

In China, grasslands are important ecosystems, accounting for more than 40% of the national land and playing important roles in servicing the ecological environment and in socio-economics (*Kang et al., 2007*; *Ren et al., 2008*). Chinese grasslands are experiencing increasing degradation due to land-use for human activities and to climate change (*Lü et al., 2011*), yet their biodiversity is still poorly known.

In this study, we aimed at investigating the influence of habitat characteristics on the structure of carabid communities in different types of grasslands in China that reflect a gradient of aridity from the most arid to the most humid: desert steppe, typical steppe and meadow steppe (*Kang et al., 2007*). For this, we considered 12 environmental variables, including five vegetational characteristics (plant biomass (PB), cover, density, height and species richness), four soil factors (bulk density, above ground litter, moisture and temperature) and three climatic factors (humidity, precipitation and temperature). In this paper, we investigated carabid responses to these factors at the regional scale and at grassland type-level by considering three assemblage characteristics: species richness, species composition and functional diversity (measured as body size and movement diversity).

## MATERIALS AND METHODS

### Study areas and sampling design

The study was carried out in the Ningxia region, northern China. We selected three sampling areas representing the three main ecosystems in the region: desert steppe, typical steppe and meadow steppe (Figs. 1 and 2). In each area, we identified different habitats to reflect within-ecosystem variability. We selected the study sites to be representative of the variability of environmental conditions within and between Chinese grassland ecosystems. We adopted a stratified sampling design, with a different number of trapping sites among grassland types to reflect their within-ecosystem variability. To make results comparable, we used the same number of traps (15) for each habitat within each ecosystem.

The desert steppe area (Fig. 2A) is located in eastern Ningxia, Yanchi county (37°59′13″N–107°05′42″E). This area is characterized by a cold, semi-arid continental monsoonal climate zone (*Liu et al., 2015*), with an average annual temperature of 8.3 °C (−8 °C in January, 22 °C in July), and average annual precipitation of around 200 mm (*Kang et al., 2007*). The vegetation is characterized by typical drought-tolerant plant species, such as *Agropyron mongolicum*, *Artemisia desertorum*, *Artemisia blepharolepi* and *Stipa* spp. The typical steppe area (Fig. 2B) is located in southern Ningxia, Guyuan County, near the Natural Reserve of the Yunwu Mountain. This area is characterized by a continental monsoon climate. Average annual temperature is 5.7 °C (−22 °C in January, 28 °C in August) and annual rainfall is 350 mm (*Kang et al., 2007*). The top of the mountain (36°12′16″N–106°24′37″E) is characterized by grass vegetation crossed by

 

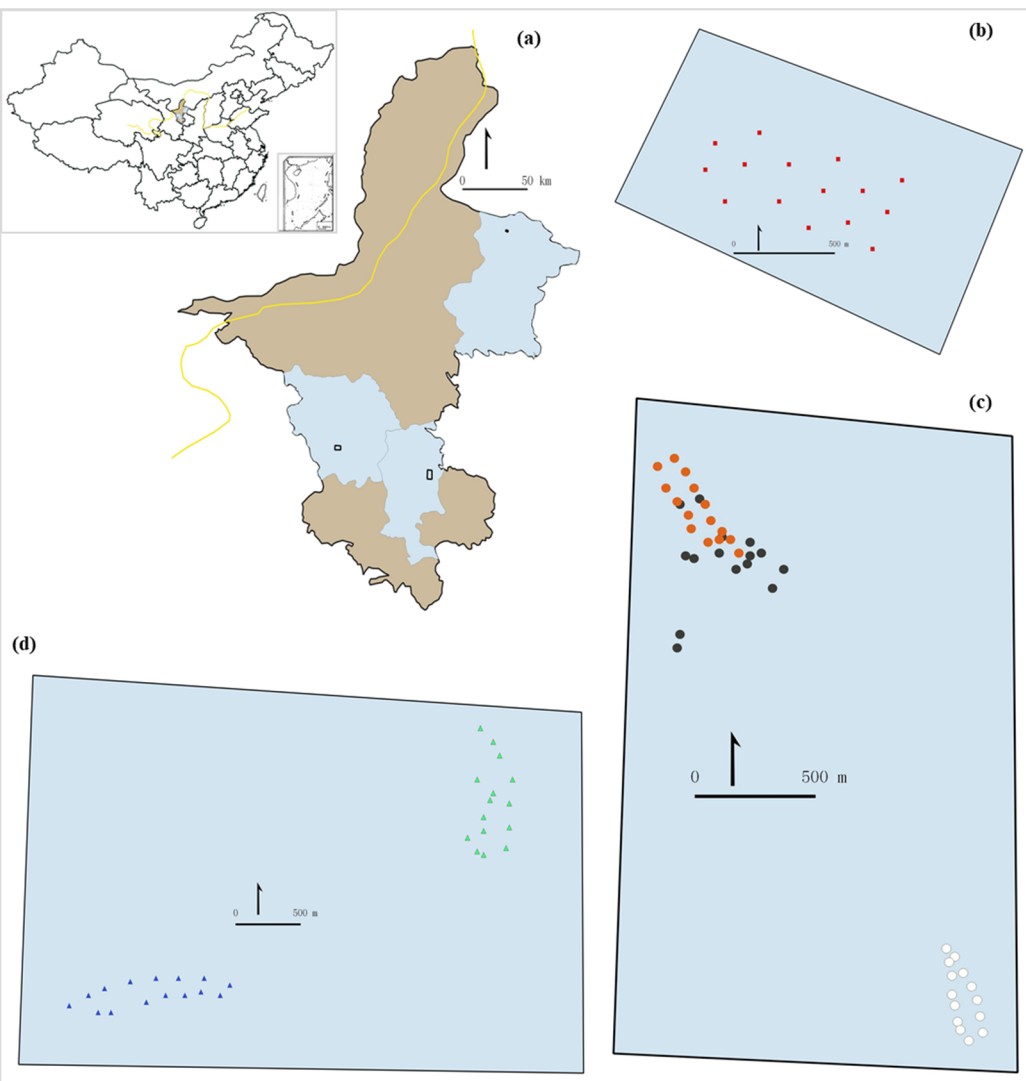

**Figure 1 Study area (A) and sampling sites (B–D).** The inset (A) shows the location of the study area in China. The light brownish colored areas represent the rest of the region contrasting the blue areas which are the counties were the grasslands were selected. (B) shows the 15 sampling points in desert steppe. (C) shows the 45 samplings points in the three sectors of the meadow steppe (orange dots: sampling points in fire belts on the mountain top; dark dots: sampling points in natural patches on the mountain top; white dots: sampling points in the mountain bottom). (D) shows the 30 sampling points in the two sectors of the meadow steppe (green dots: south-west side; blue dots: mountain bottom).

patches of cut grasses that serve as fire belts. The natural vegetation on the top of the mountain includes *Stipa bungeana*, *S. grandis*, *Artemisia frigida*, *Thymus mongolicus* and *Heteropappus altaicus*. The bottom of the mountain (36°15′6″N–106°23′5″E) is occupied by crop fields and natural vegetation, including *S. bungeana*, *Artemisia frigida*, *T. mongolicus* and *Potentilla acaulis*. In this area, we selected three sectors; the first and second sectors were located at the top of the mountain, in natural patches of grass vegetation and in fire belts, respectively; the third sector was selected at the bottom of the mountain. The meadow steppe area (Fig. 2C) is located in western Ningxia, Haiyuan

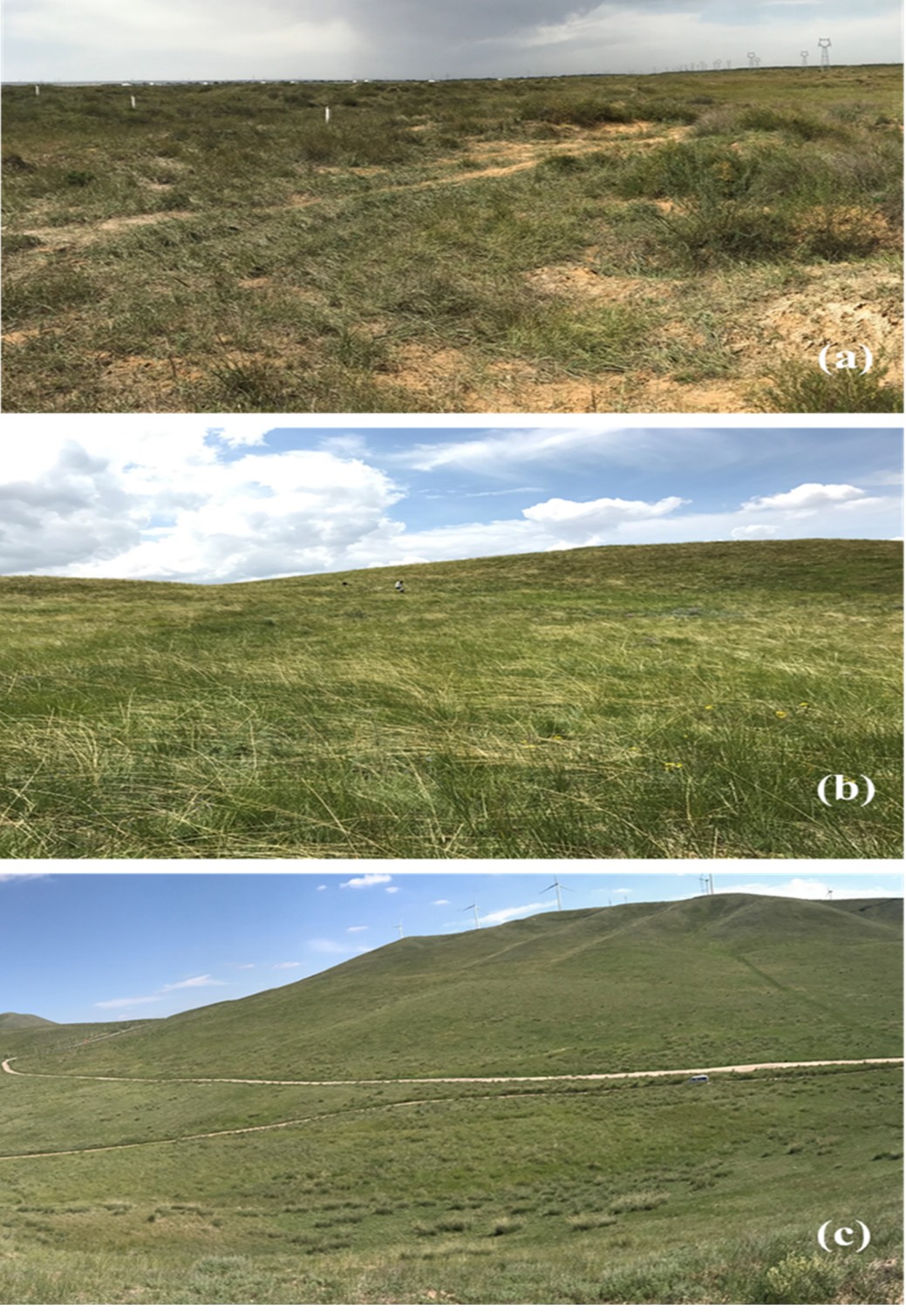

**Figure 2** **The three types of grasslands investigated in this study.** (A) Desert steppe, (B) typical steppe and (C) meadow steppe.

county, near the Nanhua mountain. This area is characterized by a semi-humid climate, with an annual temperature of 7 °C (−7 °C in January, 20 °C in July), and average annual precipitation of 450 mm (*Kang et al., 2007*). Within this area, we selected two sampling sectors to reflect different vegetation types. The first sector (36°26′50″N–105°38′24″E) was located at the south-west side of the mountain peak (2,600 m) and is dominated by several species of the genus of *Festuca*, principally the alpine fescue *Festuca brachyphylla*. The second sector (36°25′13″N–105°36′41″E) was located at the bottom of the mountain peak (1,800 m) and its vegetation is dominated by *S. bungeana*, *Artemisia frigida* and *Achnatherum splendens*.

In total, we selected 15 sampling sites in the desert steppe, 45 sites (15 sites per sector) in the typical steppe and 30 sites in the meadow steppe (15 sites per sector).

Sites were separated by at least 150 m from each other. At each sampling site, five pitfall traps were placed at a distance of least five m from each other. Pitfall traps were made of plastic cups (diameter: 7.15 cm, depth: nine cm) dug into the ground and filled with 60 ml of an attractant solution (vinegar, sugar, 70% alcohol and water in the following proportion: 2:1:1:20). We used pitfall traps with a diameter slightly over than seven cm because this size meets *Luff's (1975)* suggested optimal diameter (seven cm) for carabids. Pitfall traps were put down once a month in mid-month, from May to September 2017, and collected 3 days after. A period of 3 days was chosen because many traps were found completely full of beetles (especially Tenebrionidae) in 2 or 3 days. In total, we used 2,250 pitfall traps (90 sampling sites × 5 pitfall traps × 5 sampling dates). Prior to analyses, we pooled the data from the five pitfall traps of each sampling, because soil and vegetation characteristics were observed at the sampling site level. Trap content was sorted in the laboratory and carabids identified to species level and assigned to trophic categories (herbivores vs. predators). All material is preserved in the insect collections of the School of Agriculture of Ningxia University.

## Vegetation and soil characteristics

At each sampling site, we set up one quadrat frame of 0.25 m$^2$ to record plant dry biomass (PB, g/m$^2$), cover (PC, % of soil covered by plants), density (PD, number of plants per m$^2$), height (PH, average, cm) and species diversity, expressed as richness (PSD). Near the quadrat frames we collected samples of above-ground litter to measure litter dry mass (SL, g/m$^2$) and samples of soil (10 cm depth) to measure SM (%) and SBD (g/cm$^3$). SM was estimated using the thermogravimetric method also known as the oven dry method (*Majumdar, 2001*): SM = $[(W_2−W_3)/(W_3−W_1)] \times 100$ where $W_1$ is the weight of the empty aluminum box (g); $W_2$ is the weight of the box + soil sample (g) and $W_3$ is the weight of the box and oven dry soil (g). SBD was estimated using the ring knife method (*Bi, Zou & Zhu, 2014*), with the formula SBD = $(W_r \times 100)/(V_r \times (100 + SM))$ where $W_r$ is the weight of the soil in the ring knife; $V_r$ is the ring knife volume. We also measured ST (10 cm depth) using a portable multiparametric probe TRS-II (Zhejiang Tuopu Instrument Co. Ltd., Hangzhou City, China; accuracy of ± 0.5 °C). Monthly mean values of humidity (Hum), precipitation (Prec) and temperature (Temp) were recorded from the meteorological stations in each county.

Range, mean and standard deviation for the aforementioned environmental variables are given in Table S1.

## Species characteristics

We expressed functional diversity with reference to two aspects: dispersal power and body size. For dispersal (FD-movement), we used the following morphometric traits, under the assumption that longer and more robust legs facilitate beetle movements on the ground: (1) length (from apex to coxa) and (2) maximum width of metafemurs, (3) length of metatibiae, (4) length of metatarsi and (5) presence of wing. For body size (FD-size), we used: (1) width of the head, (2) pronotum maximum width and (3) pronotum maximum height, (4) elytral length and (5) elytral width. All these traits were used to compute a total functional diversity (FD-total). To calculate FD-total, we also considered the following traits: (1) length of antennae, (2) feeding habits (predators vs herbivores) and (3) five characteristics in the mandibles (density of ventral groove; roughness of dorsal crenulations; sharpness of incisor ridge; ratio width/length of left and right mandible). Measurements were done using a digital caliper (precision to 0.01 mm, Stainless Hardened). For species with more than 50 collected individuals, we measured 50 specimens (14 species); for other species, we measured a variable number of individuals depending on their abundance (11 species).

## Data analysis

We assessed species richness using the individual-based rarefaction method implemented in the "iNEXT" library of R (*Hsieh, Ma & Chao, 2016*). The input matrix was based on species abundance (total number of individuals from each sample site). We rarefied data to the smallest number of collected individuals. Since the number of traps in the different sites was different, in all analyses dealing with species abundance we used species' activity density, calculated as the number of individuals from each species divided by the number of traps used in each site.

For each trait, we calculated functional diversity indices using Rao's quadratic entropy, which expresses the sum of the dissimilarities in the trait space among all possible pairs of species weighted by species-relative abundances (*Rao, 1982*; *Botta-Dukát & Wilson, 2005*). High functional divergence should indicate a high degree of niche differentiation (*Mason et al., 2005*). To express functional diversity based on multiple traits, we calculated species dissimilarities with the commonly used Gower distance (*Pavoine et al., 2009*; *Laliberté & Legendre, 2010*). Calculations were done using the "StatMatch" package and the "melodic" function in R (*De Bello et al., 2016*).

Differences in species richness and functional diversity between the three grassland types were tested using a Nested analysis of variance (Nested ANOVA, with type of grasslands as fixed effect and sub-types of grassland as random effect), followed by post-hoc Tukey tests using the "multcomp" package in R (*Hothorn, Bretz & Westfall, 2008*). The effects of vegetation, soil and climate characteristics on beetles rarefied richness and functional diversity were investigated using a random-effect eigenvector spatial

filtering (RE-ESF) approach (*Murakami & Griffith, 2015*) with the "spmoran" package in R (*Murakami, 2018*) to take into account spatial dependence.

A Moran test showed a spatial autocorrelation at regional level (Moran's $I = 0.024$, $P$-value $< 0.001$) but not at the grassland scale (Desert steppe: Moran's $I = -0.008$, $P$-value $= 0.151$; Typical steppe: Moran's $I = -0.004$, $P$-value $= 0.397$; Meadow steppe: Moran's $I = -0.007$, $P$-value $= 0.645$). However, we decided to use the RE-ESF in all models to make the results comparable.

Preliminary to RE-ESF analysis, variance inflation factors (VIF) were calculated using the "usdm" package (*Naimi et al., 2014*) to detect possible collinearity between explanatory variables and determine the stability of models. A high VIF (>10) indicates that the predictor is strongly dependent on others and does not carry independent information. No collinearity was found between the variables, with all being VIF <10 (Table S2).

Using the matrix of geographical coordinates, Moran's eigenvectors and their corresponding eigenvalues were calculated using the "meigen" function in the "spmoran" package. The resulting eigenvectors are used as synthetic explanatory variables in regression analysis (*Griffith & Peres-Neto, 2006*).

The effects of vegetation, soil and climate characteristics on beetle community structure were investigated using Canonical correspondence analysis (CCA) with abundance data. This technique was particularly appropriate to our data because it addresses with the double-zero problem which characterizes community compositional data (*Legendre & Gallagher, 2001*) and does not try to display all variation in the data, but only the part that can be explained by the constraints (*Oksanen et al., 2015*). Permutation tests (999 permutations) were run to assess model significance. The sum of the canonical eigenvalues was used as a measure of the variability in the response variables explained by predictors. The importance of predictors was assessed beforehand by using the VIF (Table S3; *Oksanen et al., 2015*). Analyses were conducted in R using the "vegan" package (*Dixon, 2009*; *Oksanen et al., 2015*). We used the "step" function to determine the best model and the most important predictors in each CCA.

The variables used in the RE-ESF and CCAs were not redundant and represent different, but not mutually exclusive, hypotheses and each hypothesis has been evaluated individually with a selection procedure. We think that a further adjustment of $P$-values would result in a higher risk of pruning variables that are important. Thus, we did not adjust the $P$-values of variables selected as significant, but focused on the magnitude of the $P$-values and the consistency of results (see *Moran, 2003*).

Finally, we investigated patterns of ß-diversity, that is, species variations among habitats. We used the approach of *Baselga, Jiménez-Valverde & Niccolini (2007)* and *Baselga (2010, 2012)* for partitioning the overall ß-diversity (ßsor, Sørensen coefficient) among habitats into true species-replacement or pure turnover (ßsim, Simpson coefficient) and nestedness (ßnest = ßsor − ßsim) components. In this respect, nestedness quantified the part of compositional change caused by ordered species loss, whereas pure turnover was related to the exchange in species composition. Relationships among

**Table 1 Carabid species, their trophic group (H, herbivores; P, predators) and abundances (total number of collected beetles in brackets) in three grassland ecosystems in northern China.**

| Species name and trophic group | Species abbreviation | Regional scale (N = 6,873) | Grassland types | | |
|---|---|---|---|---|---|
| | | | Desert steppe (N = 338) | Typical steppe (N = 4,206) | Meadow steppe (N = 2,329) |
| *Amara dux* Tschitscherine, 1894. H | amar.dux | 67 | 6 | 52 | 9 |
| *Amara harpaloides* Dejean, 1828. H | amar.harp | 11 | 7 | 3 | 1 |
| *Amara helva* Tschitscherine, 1898. H | amar.helv | 9 | 9 | 0 | 0 |
| *Amara* sp. H | amara.sp | 15 | 0 | 9 | 6 |
| *Broscus kozlovi* Kryzhanovskij, 1995. P | bros.kozl | 8 | 0 | 2 | 6 |
| *Calosoma anthrax* Semenov, 1900. P | calo.anth | 41 | 0 | 34 | 7 |
| *Calosoma chinense*, Kirby, 1819. P | calo.chin | 3 | 1 | 1 | 1 |
| *Calosoma lugens* Chaudoir, 1869. P | calo.luge | 11 | 0 | 9 | 2 |
| *Carabus anchocephalus* Reitter, 1896. P | cara.anch | 85 | 0 | 29 | 56 |
| *Carabus crassesculptus* Kraatz, 1881. P | cara.cras | 339 | 0 | 0 | 339 |
| *Carabus gigoloides* Cavazzuti, 2000. P | cara.gigo | 267 | 0 | 0 | 267 |
| *Carabus glyptoterus* Fischer Von Waldheim, 1827. P | cara.glyp | 886 | 252 | 587 | 47 |
| *Carabus modestulus* Semenov, 1887. P | cara.mode | 84 | 0 | 0 | 84 |
| *Carabus sculptipennis* Chaudoir, 1877. P | cara.sculp | 404 | 0 | 401 | 3 |
| *Carabus vladimirskyi* Dejean, 1830. P | cara.vlad | 2,039 | 3 | 1,212 | 824 |
| *Corsyra fusula* Fischer Von Waldheim, 1820. H | cors.fusu | 3 | 3 | 0 | 0 |
| *Cymindis binotata* Fischer Von Waldheim, 1820. P | cymi.bino | 19 | 19 | 0 | 0 |
| *Dolichus halensis* Schaller, 1783. P | doli.hale | 3 | 0 | 3 | 0 |
| *Harpalus lumbaris* Mannerheim, 1825. H | harp.lumb | 11 | 11 | 0 | 0 |
| *Poecillus fortipes* Chaudoir, 1850. P | poec.fort | 552 | 0 | 338 | 214 |
| *Poecillus gebleri* Dejean, 1828. P | poec.gebl | 1,145 | 0 | 1,134 | 11 |
| *Pseudotaphoxenus mongolicus* (Jedlicka, 1953). P | pseu.mong | 77 | 23 | 54 | 0 |
| *Pseudotaphoxenus rugupennis* Faldermann, 1836. P | pseu.rugu | 310 | 3 | 257 | 50 |
| *Reflexisphodrus reflexipennis* Semenov, 1889. P | refle.refle | 368 | 0 | 2 | 366 |
| *Zabrus potanini* Semenov, 1889. H | zabr.pota | 116 | 1 | 79 | 36 |

assemblages were investigated by cluster analysis using the UPGMA (unweighted pair-group method, arithmetic average) amalgamation rule. Calculations were done with PAST v.3 (*Hammer, Harper & Ryan, 2001*).

# RESULTS

## Differences in richness and functional diversity

We collected a total of 6,873 individuals belonging to 25 carabid species (Table 1). Overall, 18 species were predators and seven were herbivores (six herbivores and six predators in the desert steppe, four herbivores and 14 predators in the typical steppe, and four herbivores and 15 predators in the meadow steppe). Range, mean and standard deviation of rarefied richness, FD-total, FD-movement and FD-size are given in Table S4. The desert steppe was the grassland type with the lowest values of rarefied species richness and

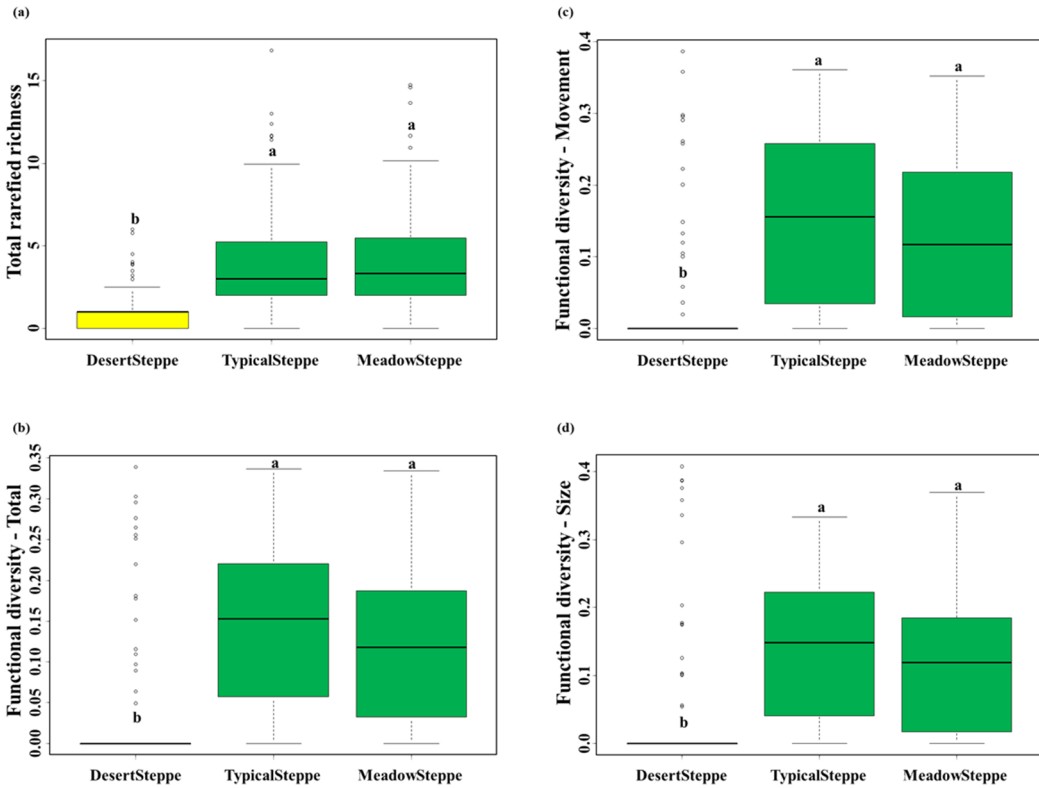

**Figure 3 Carabid community structure.** Boxplots (median, interquartile range, range and outliers) of (A) total rarefied richness, (B) total functional diversity, (C) functional diversity for movement traits and (D) functional diversity for size traits in the three investigated grassland types: desert, typical and meadow steppes. Same letter indicates non-significant differences according to Tukey tests after Nested ANOVAs. Number of sampled individuals: 6,873 at regional level, 338 in the desert steppe, 4,206 in the typical steppe and 2,329 in the meadow steppe.  

the three measures of functional diversity, whereas no significant differences were found between the meadow and the typical steppe (Fig. 3; Table S5).

## Influence of environmental variables on richness and functional diversity

Climate predictors had strong effects on carabid species richness (Table 2). Hum was positively related to species richness in the desert and typical steppes, as well as at the regional scale. Prec had a positive effect on richness at regional scale, but a negative effect in the desert steppe. Temp had a positive effect on richness at regional scale, in the meadow and in the typical steppes. At regional scale SBD, ST and SM had negative effects. SBD and ST also had negative effects in the typical steppe, and SBD had a positive effect in desert steppe, whereas none of the soil predictors were significant for the meadow steppe. None of the vegetation predictors were important in explaining species richness in the desert or in the meadow steppe. PC and PSD had positive effects in the typical steppe and at regional scale, respectively.

At regional scale (Fig. 4; Table 3), the total variance explained by CCA was 24%, with the first two axes accounting for 57% of the explained variance. Constraints were

**Table 2 Results of RE-ESF (random effect eigenvector spatial filtering) between habitat characteristics and carabid rarefied richness at regional scale and for the three grassland types separately.**

| | | Regional scale | Grassland types | | |
| --- | --- | --- | --- | --- | --- |
| | | | Desert steppe | Typical steppe | Meadow steppe |
| Model characteristics | $r^2$ | 0.35 | 0.44 | 0.32 | 0.38 |
| | rlogLik | −1,032.08 | −112.92 | −515.05 | −341.99 |
| | AIC | 2,098.15 | 259.84 | 1,064.09 | 717.98 |
| | BIC | 2,168.01 | 299.24 | 1,122.17 | 769.16 |
| Vegetation | PB | −0.28 ± 0.16 (0.083) | 0.13 ± 0.13 (0.322) | 0.27 ± 0.28 (0.338) | −0.08 ± 0.27 (0.759) |
| | PC | −0.19 ± 0.20 (0.344) | 0.30 ± 0.24 (0.210) | **0.59 ± 0.30 (0.049)** | −0.25 ± 0.26 (0.341) |
| | PD | −0.16 ± 0.18 (0.360) | −0.38 ± 0.21 (0.067) | 0.28 ± 0.26 (0.292) | 0.12 ± 0.26 (0.643) |
| | PH | −0.24 ± 0.20 (0.212) | −0.03 ± 0.20 (0.894) | −0.51 ± 0.29 (0.082) | −0.15 ± 0.38 (0.693) |
| | PSD | **0.36 ± 0.14 (0.013)** | −0.18 ± 0.17 (0.294) | 0.16 ± 0.20 (0.411) | 0.48 ± 0.24 (0.051) |
| Soil | SBD | **−0.38 ± 0.18 (0.031)** | **0.30 ± 0.14 (0.032)** | **−0.46 ± 0.20 (0.025)** | −0.51 ± 0.28 (0.072) |
| | SL | 0.26 ± 0.17 (0.129) | 0.19 ± 0.14 (0.188) | −0.22 ± 0.31 (0.470) | 0.19 ± 0.23 (0.402) |
| | SM | **−0.42 ± 0.20 (0.035)** | 0.30 ± 0.15 (0.059) | −0.12 ± 0.19 (0.528) | −0.00 ± 0.28 (0.990) |
| | ST | **−0.71 ± 0.23 (0.002)** | 0.25 ± 0.39 (0.519) | **−0.53 ± 0.20 (0.008)** | 0.07 ± 0.36 (0.841) |
| Climate | Hum | **0.54 ± 0.18 (0.003)** | **1.02 ± 0.23 (<0.0001)** | **1.19 ± 0.32 (0.0001)** | 0.21 ± 0.35 (0.546) |
| | Prec | **0.51 ± 0.16 (0.002)** | **−1.30 ± 0.36 (<0.0001)** | 0.42 ± 0.23 (0.064) | 0.26 ± 0.32 (0.423) |
| | Temp | **1.01 ± 0.17 (<0.0001)** | 0.20 ± 0.23 (0.374) | **1.37 ± 0.27 (<0.0001)** | **1.50 ± 0.27 (<0.0001)** |
| | Intercept | **3.41 ± 0.11 (<0.0001)** | **1.05 ± 0.12 (<0.0001)** | **3.87 ± 0.15 (<0.0001)** | **3.90 ± 0.19 (<0.0001)** |

Notes:
Significant effects are in bold.
Model characteristics: $r^2$, adjusted coefficient of determination; rlogLik, restricted log-likehood; AIC, Akaike information criterion; BIC, Bayesian information criterion. Parameter estimated coefficients (± standard error) and $P$-values (in parentheses) are given for each predictor. Predictors abbreviations: PB, plant dry biomass; PC, plant cover; PD, plant density; PH, plant height; PSD, plant species diversity (richness); SBD, soil bulk density; SL, soil litter; SM, soil moisture; ST, soil temperature; Hum, humidity; Prec, precipitation; Temp, temperature.

significant ($F = 9.53$, $P < 0.001$) and all predictors were retained in the selection procedure (Table S6). However, the first axis showed that SM and ST were the most important variables, acting in opposite directions. In the desert steppe community (Fig. 5; Table 3), CCA explained 40% of total variance, with the first two axes accounting for 56% of the explained variance. Constraints were not significant ($F = 1.53$, $P = 0.07$), with only humidity and temperature being included in the final model after the selection procedure (Table S6). In the typical steppe community (Fig. 6; Table 3), CCA explained 37% of total variance, with the first two axes accounting for 68% of the explained variance. Constraints were significant ($F = 9.74$, $P = 0.01$), and the final model included PB (as the most important variable), PC, PD, SL, Hum, Prec and Temp (Table S6). In the meadow steppe community (Fig. 7; Table 3), CCA explained 25% of total variance, with the first two axes accounting for 65% of the explained variance. Constraints were significant ($F = 3.38$, $P = 0.01$), with PB, PH, SL, Hum and Temp being retained in the final model. PH and Temp were the most important variables (Table S6).

The response of functional diversity to habitat characteristics (vegetation, soil and climate) differed among grassland types (Table 4). At regional scale, five predictors showed important effects on FD-total: PB (negative, marginally non-significant),

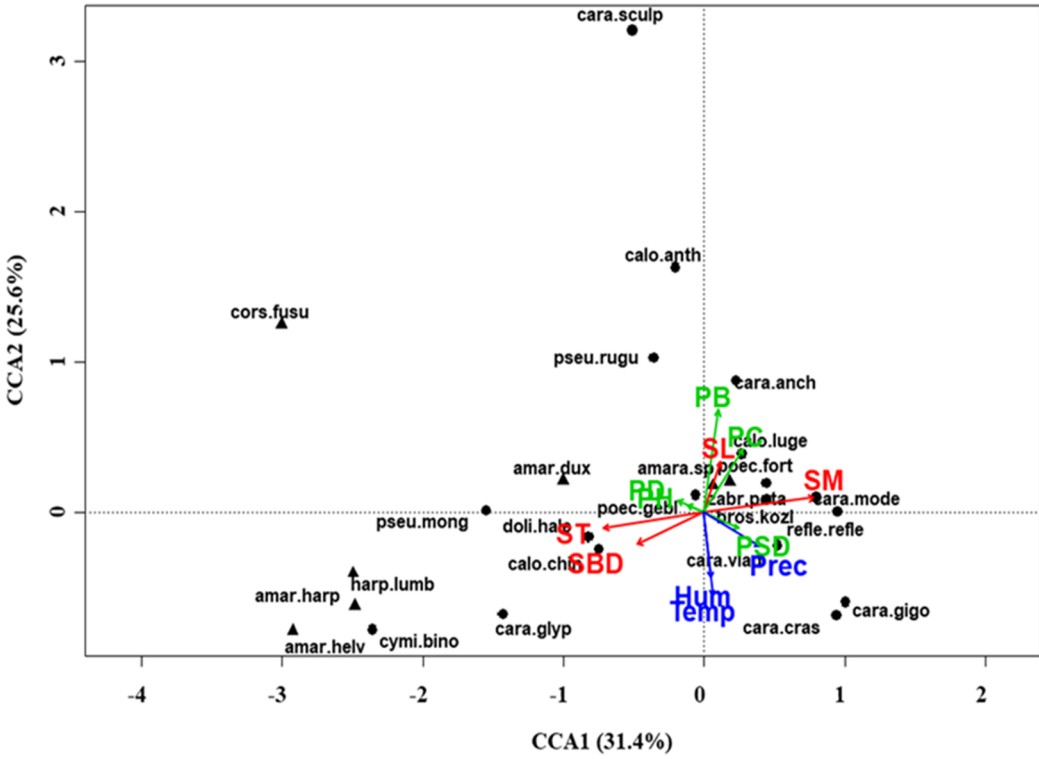

**Figure 4 Canonical correspondence analysis (CCA) biplot at regional scale.** Plot shows relationships between species (● = predators, ▲ = herbivores; species abbreviations as in Table 1) and environmental variables (vegetation, green arrows: PB, plant biomass; PC, plant cover; PD, plant density; PH, plant height; PSD, plant species diversity; soil, red arrows: SBD: bulk density; SL, soil litter; SM, soil moisture; ST, soil temperature; and climate, blue arrows; Hum, humidity; Prec, precipitation; Temp, temperature) at regional scale. See Table 1 for species names abbreviations.

ST (negative, marginally non-significant), Hum, Prec and Temp (all positive). In the desert steppe only two predictors had important effects: PD (negative, marginally non-significant) and Hum (positive). In the typical steppe, only the climate characteristics Hum and Temp were important and both had positive effects. In the meadow steppe, PC (negatively) and ST (positively) were important predictors of meadow FD-total (temperature had a marginally non-significant positive effect).

The response of FD-movement (Table S7) at regional level was similar to that of FD-total, with the exception of Hum, which was not significant. By contrast, Hum was the only important predictor of FD-movement in the desert steppe. As at the regional scale, ST, Hum and Temp were important predictors of FD-movement in the typical steppe. In the meadow steppe, PB and PC had negative effects, whereas ST and Prec had positive effects.

At regional scale, four predictors showed important effects on FD-size (Table S8): ST (negatively) and Prec, Hum and Temp (positively). In the desert steppe, two variables influenced FD-size: Hum (positively) and PD (negatively and marginally non-significant). In the typical steppe, only Hum and Temp showed significantly positive effects on FD-size. In the meadow steppe, important predictors where PC and PH (negatively), and Prec (marginally non-significant) and Temp (both positively).

**Table 3  Results of CCA (Canonical correspondence analysis).**

| | | Regional scale | | | Grassland types | | | | | | | | |
|---|---|---|---|---|---|---|---|---|---|---|---|---|---|
| | | | | | Desert steppe | | | Typical steppe | | | Meadow steppe | | |
| Eigenvalues | Total constrained (proportion %) | 1.86 (23.66%) | | | 1.68 (39.65%) | | | 1.22 (37.48%) | | | 1.13 (25.26%) | | |
| | CCA1 | 0.58 (31.43; 0.001) | | | 0.54 (26.96; 0.246) | | | 0.56 (46.15; 0.001) | | | 0.49 (43.76; 0.001) | | |
| | CCA2 | 0.48 (25.63; 0.001) | | | 0.49 (29.34; 0.241) | | | 0.27 (21.85; 0.001) | | | 0.24 (21.19; 0.001) | | |
| | CCA3 | 0.33 (17.56; 0.001) | | | 0.30 (17.79; 0.704) | | | 0.22 (17.66; 0.097) | | | 0.16 (14.59; 0.002) | | |
| Biplot scores for constraining variables | | CCA1 | CCA2 | CCA3 | CCA1 | CCA2 | CCA3 | CCA1 | CCA2 | CCA3 | CCA1 | CCA2 | CCA3 |
| Vegetation | PB | 0.103 | 0.675 | −0.027 | −0.062 | 0.039 | −0.599 | 0.708 | −0.109 | 0.008 | −0.020 | 0.195 | 0.396 |
| | PC | 0.280 | 0.413 | −0.531 | 0.368 | −0.334 | −0.257 | 0.115 | −0.618 | 0.318 | 0.028 | 0.310 | 0.245 |
| | PD | −0.181 | 0.081 | −0.763 | 0.105 | −0.490 | −0.192 | −0.352 | −0.511 | −0.090 | 0.089 | 0.332 | 0.429 |
| | PH | −0.122 | 0.045 | −0.774 | 0.192 | −0.414 | −0.278 | −0.400 | −0.583 | 0.160 | 0.697 | −0.097 | 0.624 |
| | PSD | 0.249 | −0.109 | −0.009 | 0.372 | −0.482 | 0.189 | −0.167 | 0.062 | −0.124 | 0.309 | −0.152 | −0.104 |
| | SBD | −0.471 | −0.213 | 0.279 | −0.089 | 0.093 | −0.296 | 0.073 | 0.271 | 0.004 | −0.311 | −0.139 | −0.317 |
| Soil | SL | 0.121 | 0.333 | −0.234 | 0.148 | 0.285 | −0.068 | 0.168 | −0.470 | 0.182 | 0.156 | −0.306 | −0.357 |
| | SM | 0.779 | 0.094 | 0.096 | 0.092 | 0.188 | −0.187 | −0.037 | −0.317 | 0.004 | −0.151 | 0.275 | −0.102 |
| | ST | −0.772 | −0.104 | −0.487 | −0.022 | −0.147 | 0.003 | −0.487 | 0.143 | 0.005 | 0.482 | −0.132 | 0.391 |
| Climate | Hum | 0.048 | −0.436 | −0.025 | 0.663 | 0.193 | 0.229 | −0.455 | −0.290 | −0.742 | −0.324 | 0.412 | 0.708 |
| | Prec | 0.402 | −0.226 | −0.279 | 0.019 | −0.417 | −0.129 | −0.431 | −0.207 | −0.305 | −0.340 | 0.126 | 0.749 |
| | Temp | 0.059 | −0.554 | −0.163 | 0.010 | −0.439 | −0.108 | −0.633 | 0.295 | 0.562 | −0.130 | −0.913 | −0.051 |

Notes:
Percentages of variance explained and *P*-values are given in parentheses.
Predictor abbreviations: PB, plant dry biomass; PC, plant cover; PD, plant density; PH, plant height; PSD, plant species diversity (richness); SBD, soil bulk density; SL, soil litter; SM, soil moisture; ST, soil temperature; Hum, humidity; Prec, precipitation; Temp, temperature.

## Beta diversity

Species composition varied greatly among grassland types, with seven species shared by all three types (Fig. 8). The overall ß-diversity (ßsor) pattern (Fig. 9A) indicated that the desert steppe differed the most. The meadow sector occupied by *Festuca* spp. also clustered apart, whereas the other meadow sector, occupied by *Stipa* spp., clustered with the typical steppe. When the pure turnover (ßsim) is considered (Fig. 9B), the desert was again identified as the ecosystem differing the most. All three sectors of the typical steppe clustered together and were separated by the two meadow sectors. In the analysis of the nestedness component (ßnest) the desert steppe clustered with the sector of typical steppe located at the bottom of the mountain peak and the meadow steppe dominated by *Festuca* spp., whereas the other meadow sector clustered with the typical steppe (Fig. 9C).

## DISCUSSION

We found positive relationships between carabid richness and temperature. Temperature is known as a major predictor of species richness for several organisms (*Allen, Brown & Gillooly, 2002*; *Peters et al., 2016*; *Yu et al., 2016*) and is an important factor in the life-cycle of most carabid species (*Thiele, 1977*). The lack of any influence of temperature

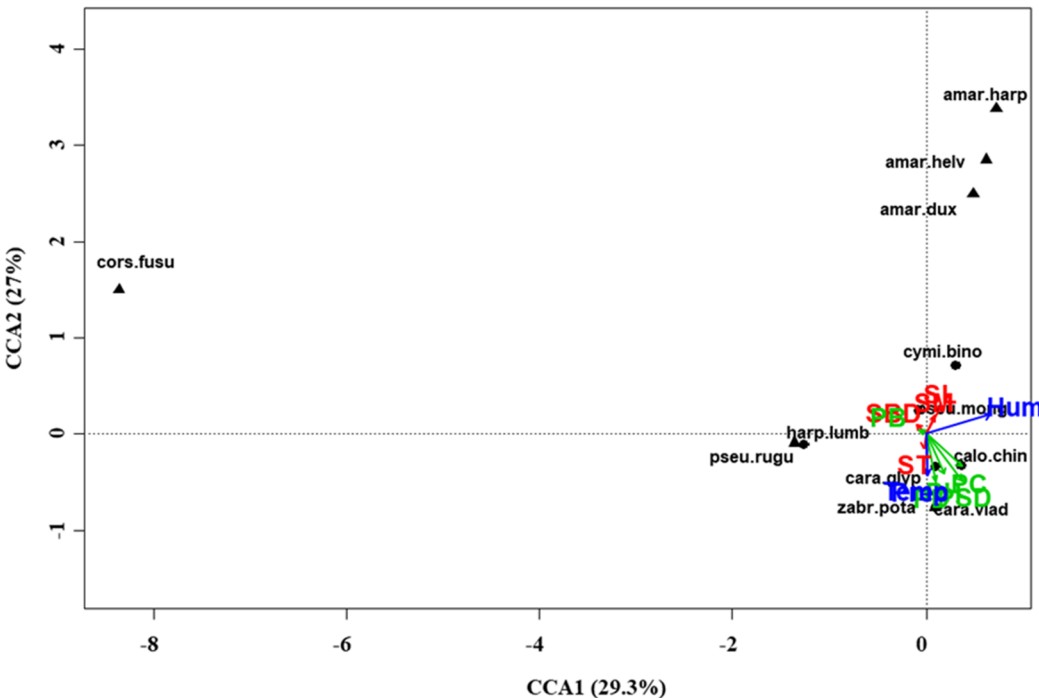

**Figure 5 Canonical correspondence analysis (CCA) biplot in the desert steppe.** Plot shows relationships between species (● = predators, ▲ = herbivores; species abbreviations as in Table 1) and environmental variables (vegetation, green arrows: PB, plant biomass; PC, plant cover; PD, plant density; PH, plant height; PSD, plant species diversity; soil, red arrows: SBD: bulk density; SL, soil litter; SM, soil moisture; ST, soil temperature; and climate, blue arrows; Hum, humidity; Prec, precipitation; Temp, temperature) in the desert steppe. See Table 1 for species names abbreviations.

on carabid richness in the desert steppe suggests that in this environment higher temperatures can be intolerable for most species. Moreover, in this environment, differences between day and night temperatures are much more pronounced than in the other grassland types, a source of variation which is not included in our measurements because this type of datum was unfortunately not available, but which might be important for carabids.

Rainfall is also a major predictor of species richness at the regional scale and in the desert, where, however, it has a negative effect. Rainfall drives many ecological processes and may influence shelter sites and food resources used by carabids (*Morecroft et al., 2004*). The negative effect of rainfall observed in the desert steppe may be due to two causes. First, it is possible that this environment hosts species that are particularly adapted to arid conditions, and are therefore negatively affected by rainfall. Second, our sampling area in the arid steppe is surrounded by many industries, and *Thiele (1977)* mentioned that carabids are highly vulnerable to polluted rainfall. Thus, it is possible that rainfall was locally polluted by industry emissions that negatively affected carabids richness. We do not have data to test this hypothesis, however.

We found a strong positive effect of humidity on carabid species richness at regional scale in desert and typical steppes, but not in the meadow steppe. This lack of

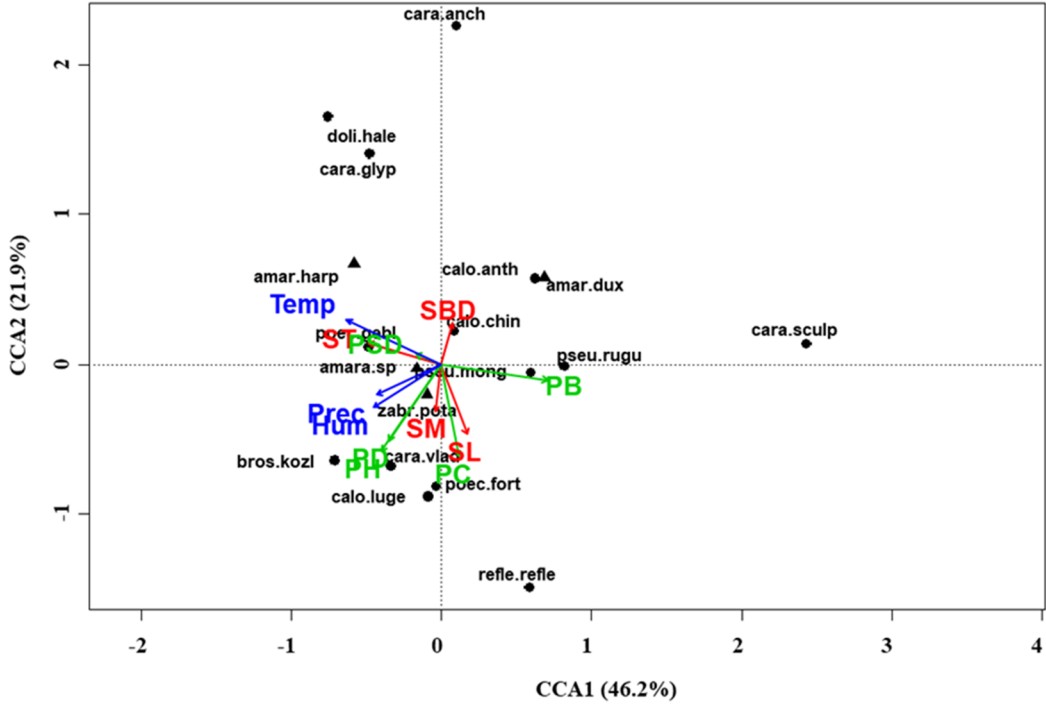

**Figure 6 Canonical correspondence analysis (CCA) biplot in the typical steppe.** Plot shows relationships between species (● = predators, ▲ = herbivores; species abbreviations as in Table 1) and environmental variables (vegetation, green arrows: PB, plant biomass; PC, plant cover; PD, plant density; PH, plant height; PSD, plant species diversity; soil, red arrows: SBD: bulk density; SL, soil litter; SM, soil moisture; ST, soil temperature; and climate, blue arrows; Hum, humidity; Prec, precipitation; Temp, temperature) in the typical steppe. See Table 1 for species names abbreviations.

influence of humidity in this ecosystem may reflect the fact that it is the most humid. Vegetation characteristics are positively related to carabid species richness at regional scale (as species richness) and in the typical steppe (as PC), but not in the desert and meadow steppes. *Rahman et al. (2015)* hypothesized that vegetation cover might accelerate the establishment of carabid communities because it provides living space and modifies the microclimate to create a heterogeneous and stratified microenvironment supporting different carabid species. The influence of plant diversity on carabid richness at the regional level, which includes a variety of habitat types with different communities, is consistent with previous studies reporting that a higher plant species richness implies more diverse food resources, thus allowing the presence of species with different feeding preferences (*Byers et al., 2000*; *Brose, 2003*). However, most carabid species are predators, so they can be influenced by plant diversity only indirectly (e.g., if a higher plant diversity supports a higher diversity of prey). In fact, several studies report a lack of significant relationships or even negative correlations between plant and arthropod diversity (see *Zou et al., 2013*). Increased plant diversity may actually promote an increase in herbivores; a higher diversity and/or abundance of herbivores, however, can represent an increase in food sources and niches for other predators (such as spiders), thus increasing the overall competition levels, and consequently reducing the overall diversity of

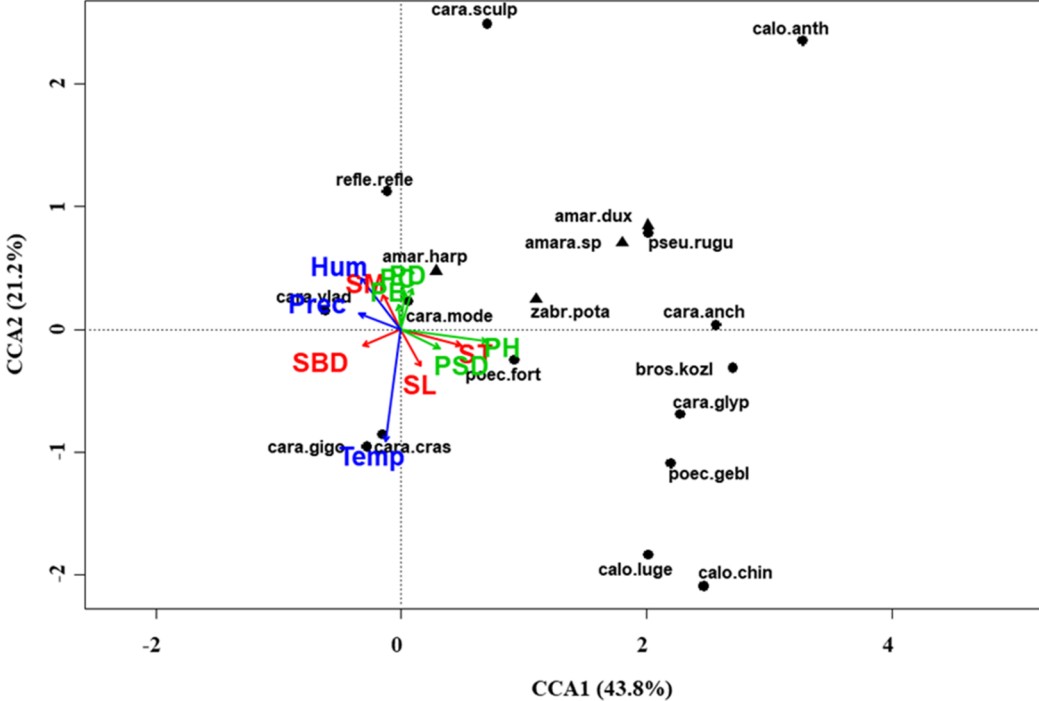

**Figure 7 Canonical correspondence analysis (CCA) biplot in the meadow steppe.** Plot shows relationships between species (● = predators, ▲ = herbivores; species abbreviations as in Table 1) and environmental variables (vegetation, green arrows: PB, plant biomass; PC, plant cover; PD, plant density; PH, plant height; PSD, plant species diversity; soil, red arrows: SBD: bulk density; SL, soil litter; SM, soil moisture; ST, soil temperature; and climate, blue arrows; Hum: humidity; Prec: precipitation; Temp: temperature) in the meadow steppe. See Table 1 for species names abbreviations.

predators. This may explain the lack of positive effects of plant diversity on carabid richness when the three ecosystems are analyzed separately.

Plant height did not influence carabid species richness. PH might enhance arthropod diversity by adding vertical strata in species niches, thus decreasing overall competition. The lack of significant effects suggests that this is not important for carabids, probably because most of them are ground-dwelling insects that do not separate their niches using a differential vertical distribution. As regards soil characteristics, bulk density (which increases with soil compaction) was negatively related to species richness at the regional scale and in the typical steppe, which may be due to the fact that soil compaction makes egg-laying and burrowing difficult (see *Magura, Tóthmérész & Elek, 2003*), but was related positively in the desert, where soil is very loose and compaction may reflect the presence of vegetation spots, which may attract carabids (e.g., by providing water and shadow). Soil surface temperature was negatively related to species richness at the regional scale, which may be explained by the fact that the highest temperatures recorded in our study system are those of the desert steppe, a grassland type with overall low carabid richness. Moisture impacted carabid richness negatively at the regional scale, which suggests that most of the species occurring in the study area are not hygrophilous but adapted to arid conditions.

**Table 4 Results of RE-ESF (random effect eigenvector spatial filtering) between habitat characteristics and carabid total functional diversity (FD-total) at regional scale and for the three grassland types separately.**

| | | Regional scale | Grassland types | | |
| --- | --- | --- | --- | --- | --- |
| | | | Desert steppe | Typical steppe | Meadow steppe |
| Model characteristics | $r^2$ | 0.28 | 0.24 | 0.21 | 0.38 |
| | rlogLik | 418.57 | 47.33 | 193.17 | 133.22 |
| | AIC | −803.13 | −60.66 | −352.34 | −232.44 |
| | BIC | −733.27 | −21.27 | −294.26 | −181.26 |
| Vegetation | PB | −0.01 ± 0.01 (0.051) | 0.01 ± 0.01 (0.307) | 0.01 ± 0.01 (0.514) | −0.01 ± 0.01 (0.200) |
| | PC | −0.01 ± 0.01 (0.083) | 0.01 ± 0.02 (0.546) | 0.01 ± 0.01 (0.443) | **−0.02 ± 0.01 (0.023)** |
| | PD | −0.01 ± 0.01 (0.417) | −0.03 ± 0.02 (0.061) | 0.01 ± 0.01 (0.170) | 0.01 ± 0.01 (0.208) |
| | PH | 0.01 ± 0.01 (0.249) | 0.01 ± 0.01 (0.708) | 0.00 ± 0.01 (0.985) | 0.02 ± 0.01 (0.070) |
| | PSD | 0.00 ± 0.01 (0.391) | −0.01 ± 0.01 (0.279) | −0.00 ± 0.01 (0.550) | 0.01 ± 0.01 (0.203) |
| Soil | SBD | −0.00 ± 0.01 (0.504) | 0.01 ± 0.01 (0.201) | −0.01 ± 0.01 (0.226) | −0.00 ± 0.01 (0.811) |
| | SL | 0.00 ± 0.01 (0.910) | 0.02 ± 0.01 (0.115) | −0.01 ± 0.01 (0.355) | 0.01 ± 0.01 (0.422) |
| | SM | −0.01 ± 0.01 (0.153) | 0.01 ± 0.01 (0.356) | −0.00 ± 0.01 (0.707) | 0.01 ± 0.01 (0.457) |
| | ST | −0.02 ± 0.01 (0.060) | −0.01 ± 0.03 (0.727) | −0.01 ± 0.01 (0.107) | **0.03 ± 0.01 (0.025)** |
| Climate | Hum | **0.02 ± 0.01 (0.005)** | **0.05 ± 0.02 (0.009)** | **0.04 ± 0.01 (0.001)** | −0.00 ± 0.01 (0.694) |
| | Prec | **0.01 ± 0.01 (0.032)** | −0.05 ± 0.03 (0.058) | 0.01 ± 0.01 (0.489) | 0.02 ± 0.01 (0.061) |
| | Temp | **0.02 ± 0.01 (<0.0001)** | 0.02 ± 0.02 (0.303) | **0.04 ± 0.01 (<0.0001)** | 0.02 ± 0.01 (0.052) |
| | Intercept | **0.12 ± 0.00 (<0.0001)** | **0.04 ± 0.01 (<0.0001)** | **0.14 ± 0.01 (<0.0001)** | **0.12 ± 0.01 (<0.0001)** |

Notes:
Parameter estimated coefficients (± standard error) and *P*-values (in parentheses) are given for each predictor. Significant effects are in bold.
Model characteristics: $r^2$, adjusted coefficient of determination; rlogLik, restricted log-likelihood; AIC, Akaike information criterion; BIC, Bayesian information criterion.
Predictors abbreviations: PB, plant dry biomass; PC, plant cover; PD, plant density; PH, plant height; PSD, plant species diversity (richness); SBD, soil bulk density; SL, soil litter; SM, soil moisture; ST, soil temperature; Hum, humidity; Prec, precipitation; Temp, temperature.

We found that climatic factors, and in particular temperature, were the most important variables in predicting the variability in carabid species composition, both at regional scale and for grassland types, except for the desert steppe, for which these factors had limited explanatory power. Temperature has been reported as the most important environmental factor for carabid communities (*Eyre et al., 2005*; *Ernst & Buddle, 2015*; *Yu et al., 2016*), and our results support this conclusion. Previous research found that vegetation and soil characteristics are also important drivers of carabid species composition (*Holmes, Boyce & Reed, 1993*; *Perner & Malt, 2003*; *Schaffers et al., 2008*; *Gioria et al., 2010*; *Birkhofer et al., 2015*; *Liu et al., 2016*; *Vogels et al., 2017*; *Ng et al., 2018b*). We found that at regional scale all vegetation and soil factors were important predictors of carabid community composition. In particular, we found that herbivorous species at the regional scale tend to be positively influenced by ST, a possible consequence of their smaller size (see *Tseng et al., 2018*).

On the other hand, carabid communities of different grassland types are influenced by different vegetation and soil characteristics. Some vegetation and soil characteristics are important for the typical and meadow steppe carabids, but not for the desert steppe community. These results indicate that different grassland types host different carabid communities that are diversely influenced by different vegetation and soil characteristics,
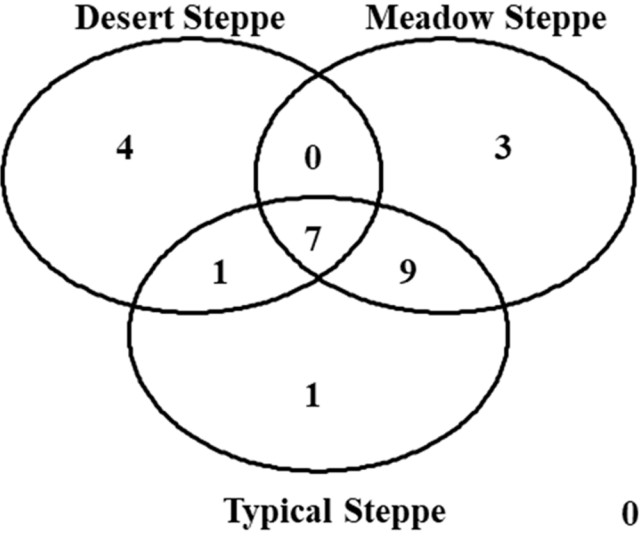

**Figure 8 Venn diagram.** The diagram shows the number of species per type of grassland.

and that virtually all of them concur to generate the overall patterns that emerge when the three grassland types are considered simultaneously.

Plant height has been reported as an important factor influencing carabid assemblages in peatlands (*Holmes, Boyce & Reed, 1993*), although the possible influence of PB remains unknown. In our study, PB and PH were the only important vegetational characteristics for the meadow carabid community, whereas all vegetational characteristics except PH were important for the typical steppe. Thus, PB was the common best vegetation predictor for both these grassland types, suggesting that ecosystem productivity (for which PB may be a proxy) is an important driver of carabid assemblage composition.

Within soil characteristics, above-ground litter was the only important predictor of species composition in the typical and meadow steppes. Previously, evidence of the role of the amount of litter in carabid species composition has been reported in forests (*Magura, Tóthmérész & Elek, 2003*; *Vician et al., 2018*) and our study indicates that this soil characteristic may be important in other ecosystems too.

None of the vegetation characteristics had an important influence in the carabid species composition of the desert steppe, even though half of the species collected in this grassland type were herbivores. These results contrast with a previous study in an arid region of the northwestern China (*Liu et al., 2016*), where shrub height and cover were important predictors of predator species composition, and shrub cover and herbaceous species richness were important predictors of herbivorous species richness. We can hypothesize that the lack of influence of any vegetation and soil characteristics on the carabid composition in our desert steppe may reflect the fact that most of these parameters have relatively similar values (as shown by their relatively small ranges and standard deviations) all through the sampling sites, compared to the typical and meadow steppes. In other words, the desert steppe was environmentally quite homogeneous, and thus there was too little variation in vegetation and soil

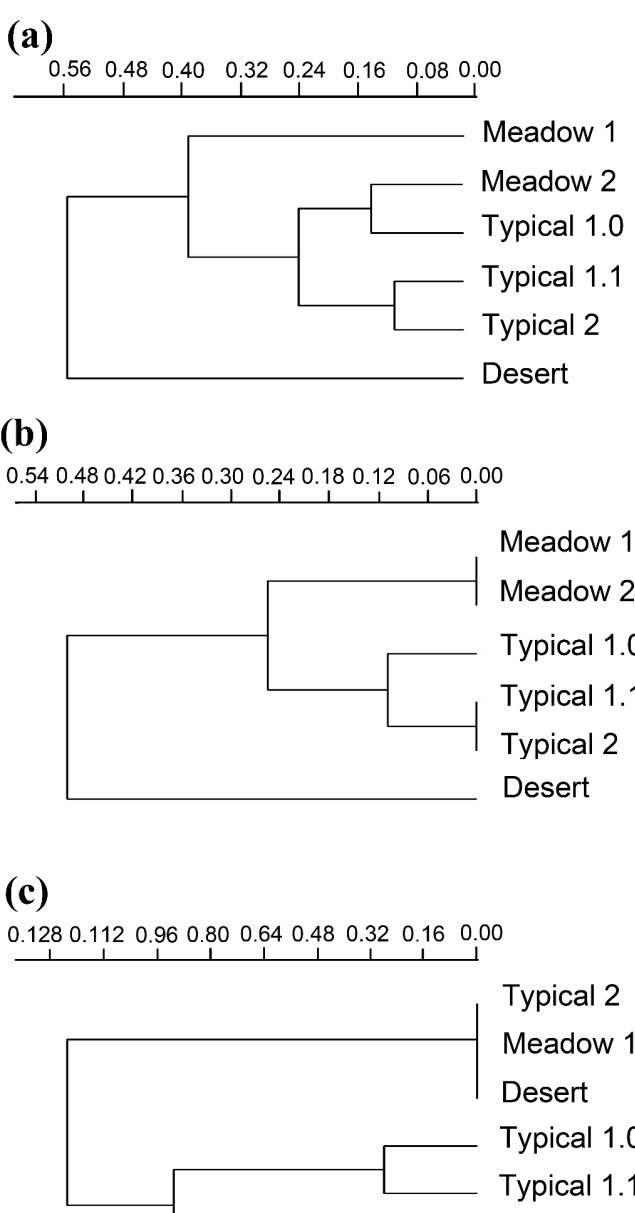

**Figure 9 Beta diversity.** Relationships between carabid communities of different grassland habitats based on ßsor (A), ßsim (B) and ßnest (C) coefficients and UPGMA clustering. Desert, desert steppe; Meadow1, top sector of the meadow steppe; Meadow2, down sector of the meadow steppe; Typical 1.0, typical steppe without fire belt; Typical 1.1, typical steppe with fire belts; Typical 2, typical steppe at the mountain bottom.

characteristics among sampling sites to generate assemblages that could be differently influenced by these factors.

All three climate factors were important variables in explaining overall functional diversity at the regional scale. Temperature was also important in the typical and in the meadow steppes, but not in the desert steppe, whereas humidity was important in the desert and the typical steppe, but not in the meadow steppe. These differences may be

attributed to the different physiological needs of the different species pools in each grassland type. Even though they are close in term of climate factors, meadow and typical steppes harbor different functional species pools which respond differently to climate factors. To our knowledge, this study is the first to investigate the relation between carabid functional diversity and climate factors in grasslands.

Looking at the different components of functional diversity in the desert steppe, humidity positively influenced both FD-movement and FD-size. This suggests that higher humidity values tend to select large-sized and more mobile species, possibly because an increase in humidity is related to soft soil conditions, higher food resources and diverse shelter and hibernation places necessary for large species.

Plant biomass had a negative effect on FD-total and FD-movement at regional scale. PC also negatively influenced FD-total and FD-movement in the meadow steppe (where PB also had a negative influence). Thus, less mobile species were associated with a higher amount of biomass. It is possible that higher biomass produces a higher quantity of debris, which hinders the movement of species. Interestingly, PD seems to have a negative effect on FD-total and FD-size in the desert steppe. Previous research reported that small body size is a characteristic of carabid beetles that inhabit severe environments, probably because of depauperate food availability (*Blake et al., 1994*; *Lövei & Magura, 2006*; *Hiramatsu & Usio, 2018*), which may explain the prevalence of small sized species in sites with low vegetation cover and density.

Soil temperature had a negative effect on all aspects of FD at the regional level, but a positive effect in the meadow carabids. A general trend is that warmer soils tend to select small-sized species (*Tseng et al., 2018*), but in soft soil, like in the meadow steppe, the rise of ST may favor both larger and highly mobile species. This may be attributed to the fact that temperature moderates predation effects (*Chase, 1996*). Thus, larger and more mobile species will be selected at the expense of small-sized species which compete less successfully during predation.

## CONCLUSIONS

Our study indicates that carabid community structure and functioning in grasslands are strongly influenced by climatic factors, and can therefore be particular sensitive to ongoing climate change. We found, however, that the responses of carabid communities to climate and other factors vary according to the grassland type, which warns against generalizations. Carabid responses to vegetation and soil characteristics also varied among grassland types, which indicates that management programs should be considered at grassland scale. Local habitat characteristics of the desert steppe seem to act as a strong filter on carabid species, allowing the presence of relatively few species and a low functional diversity. Given the currently increasing aridification processes, we can hypothesize that in the future carabid communities will be progressively more similar to those of the desert steppe, reinforcing the urgent need to implement conservation policies.

## ACKNOWLEDGEMENTS

We thank the private landowners for allowing us to place the pitfall traps on their lands. We are grateful to Professor Hongbin Liang (Institute of Zoology, CAS) for identifying specimens. We are grateful to numerous students for their help during field work and trait measurements for functional diversity, and especially to Master Zhao Yuchen. We are grateful to Jonathan Taglione, Mauro Gobbi and two anonymous referees for their comments on an early version of the manuscript.

### Funding

This work was supported by the National Natural Science Foundation of China (No. 31660630) and the first-class discipline of Practaculture Science of Ningxia University (No. NXYLXK2017A01). The funders had no role in study design, data collection and analysis, decision to publish, or preparation of the manuscript.

### Grant Disclosures

The following grant information was disclosed by the authors:
National Natural Science Foundation of China: 31660630.
Practaculture Science of Ningxia University: NXYLXK2017A01.

### Competing Interests

The authors declare that they have no competing interests.

### Author Contributions

- Noelline Tsafack conceived and designed the experiments, performed the experiments, analyzed the data, contributed reagents/materials/analysis tools, prepared figures and/or tables, authored or reviewed drafts of the paper, approved the final draft.
- François Rebaudo analyzed the data, contributed reagents/materials/analysis tools, prepared figures and/or tables, authored or reviewed drafts of the paper, approved the final draft.
- Hui Wang conceived and designed the experiments, performed the experiments, contributed reagents/materials/analysis tools, authored or reviewed drafts of the paper, approved the final draft.
- Dávid D. Nagy analyzed the data, authored or reviewed drafts of the paper, approved the final draft.
- Yingzhong Xie conceived and designed the experiments, contributed reagents/materials/analysis tools, authored or reviewed drafts of the paper, approved the final draft.
- Xinpu Wang conceived and designed the experiments, performed the experiments, contributed reagents/materials/analysis tools, authored or reviewed drafts of the paper, approved the final draft.
- Simone Fattorini analyzed the data, prepared figures and/or tables, authored or reviewed drafts of the paper, approved the final draft.

## Data Availability

Tsafack, Noelline, 2018, "Data_Carabidae in China Grasslands," DOI 10.7910/DVN/FKBCRQ, Harvard Dataverse, V1.

## Supplemental Information

Supplemental information for this article can be found online at http://dx.doi.org/10.7717/peerj.6197#supplemental-information.

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
