# Peer review of "Carabid community structure in northern China grassland ecosystems: Effects of local habitat on species richness, species composition and functional diversity"

_PeerJ, doi:10.7717/peerj.6197_

## Round 0.1 · original submission · Major Revisions

Two of the reviewers have included files with annotations. I note some concerns with grammar and appropriate statistical analyses. Please pay particular attention to these comments in your revision.

·

Basic reporting

'no comments'

Experimental design

'no comments'

Validity of the findings

'no comments'

Additional comments

Dear Authors,
I read the paper with great interest.
It deals with carabid community structure and functioning in relation to grassland types.
The most original and interesting result is that carabids are strongly influenced by local climatic factors, and the responses of carabid communities to climate and other factors vary according to the grassland type.
I appreciated the analytical approach in considering the carabid community by structural and functional approach.
I added few comments directly into the pdf.
Good luck!

Reviewer 2 ·

Basic reporting

Technically, this submission meets the standards of PeerJ. The paper is well written and the language acceptable. The literature is reasonably cited. Some of the figures can go into an electronic appendix. Figure labels are too small and hard to read. The necessary raw data are supplied.

Experimental design

This paper reports original scientific work that meets the scope of PeerJ. However, I have reservations regarding the research questions. This is another paper that starts with global change and end in a classic faunistic study. In fact it reports niche occupancy and changes of ground beetle communities across three habitat types. Numerous studies have reported on that and this paper adds another case study. This has nothing to do with global change. Thus I cannot see the gap in knowledge filled by this paper (except the local faunistic analysis). I advise a respective rewriting of the introduction. Particularly, the starting hypotheses seem to be ad hoc. They are quite obvious and reflect what we already know. Some hypotheses rather serve as strawman for the discussion. This makes the study confirmatory and descriptive.

I have also some reservations against the methods. Pitfall samples of ground beetles are common and well introduced. However, the limitations are also known. For instance, too small pitfall sizes heavily bias the results. I expected mentioning this problem. Further, this study is a classic case of pseudorreplication. Three habitats have been studied and the pitfall data within these three habitats are not independent. Thus CCA and multiple regression results are also biased (too low standard errors). There are several ways to reduce this bias. Most often used are eigenvector maps. In the present case you might also use a generalized linear mixed model with site identity as categorical random variable or the plot distances as metric covariate. As it stands the multiple regression results might be misleading. Further, the high number of variables and comparisons open the door for false positives. This should have been corrected for.
I’m also concerned about the rarefaction approach. Rarefaction is a method of standardization of sample data. In your case you extrapolate to estimated richness. Thus I suspect you have used one of the Chao estimators. This should have been stated clearly.

Validity of the findings

Given the methodological uncertainties, the discussion is rather speculative.
The introduction highlights species composition and functional aspects. However, major part of the results and discussion is on richness only. Compositional diversity and functional aspects have not been studied sufficiently. For instance, what about beta diversity, niche overlap and variability, proportional changes in abundance, small scale heterogeneity, etc.?
The discussion only marginally reflects the global change attitude of the introduction.

Unfortunately, the conclusions are quite obvious and well documented by numerous other studies. I think this study is suited (in a revised version) for a specialized carabidological journal.

Reviewer 3 ·

Basic reporting

see attached document

Experimental design

see attached document

Validity of the findings

see attached document

Additional comments

see attached document

Annotated reviews are not available for download in order to protect the identity of reviewers who chose to remain anonymous.

---

## Round 0.2 · accepted · Accept

Thank you for your efforts in revising your manuscript according to reviewer comments. I appreciate your thorough responses to those comments as well.

#